# Chondroitin Sulfate Proteoglycan 4 Provides New Treatment Approach to Preventing Peritoneal Dissemination in Ovarian Cancer

**DOI:** 10.3390/ijms25031626

**Published:** 2024-01-28

**Authors:** Kaname Uno, Yoshihiro Koya, Masato Yoshihara, Shohei Iyoshi, Kazuhisa Kitami, Mai Sugiyama, Emiri Miyamoto, Kazumasa Mogi, Hiroki Fujimoto, Yoshihiko Yamakita, Xinhui Wang, Akihiro Nawa, Hiroaki Kajiyama

**Affiliations:** 1Department of Obstetrics and Gynecology, Nagoya University Graduate School of Medicine, Nagoya 466-8560, Aichi, Japan; uno.kaname@med.nagoya-u.ac.jp (K.U.); iyoshi.shohei@med.nagoya-u.ac.jp (S.I.); kitami.kazuhisa@med.nagoya-u.ac.jp (K.K.); miyamoto.emiri@med.nagoya-u.ac.jp (E.M.); mogi.kazumasa@med.nagoya-u.ac.jp (K.M.); fujimoto.hiroki@med.nagoya-u.ac.jp (H.F.); yoshihiko-yamakita@kishokai.or.jp (Y.Y.); kajiyama@med.nagoya-u.ac.jp (H.K.); 2Division of Clinical Genetics, Department of Laboratory Medicine, Lund University Graduate School of Medicine, 22184 Lund Postcode City, Sweden; 3Bell Research Center, Department of Obstetrics and Gynecology Collaborative Research, Nagoya University Graduate School of Medicine, Nagoya 466-8550, Aichi, Japan; mai-sugiyama@kishokai.or.jp (M.S.); akihiro-nawa@kishokai.or.jp (A.N.); 4Bell Research Center for Reproductive Health and Cancer, Medical Corporation Kishokai, Nagoya 466-8550, Aichi, Japan; 5Spemann Graduate School of Biology and Medicine, University of Freiburg, 79104 Freiburg, Germany; 6Institute for Advanced Research, Nagoya University, Furo-cho, Chikusa-ku, Nagoya 464-8601, Aichi, Japan; 7Department of Obstetrics and Gynecology, Kitasato University School of Medicine, Sagamihara 252-0375, Kanagawa, Japan; 8Discipline of Obstetrics and Gynecology, Adelaide Medical School, Robinson Research Institute, University of Adelaide, Adelaide 5000, Australia; 9Department of Surgery, Massachusetts General Hospital, Harvard Medical School, Boston, MA 02114, USA; xwang30@mgh.harvard.edu

**Keywords:** ovarian cancer, peritoneal metastasis, cancer spheroids, chondroitin sulfate, cell–cell interaction

## Abstract

Most epithelial ovarian cancer (EOC) patients are diagnosed with peritoneal dissemination. Cellular interactions are an important aspect of EOC cells when they detach from the primary site of the ovary. However, the mechanism remains underexplored. Our study aimed to reveal the role of chondroitin sulfate proteoglycan 4 (CSPG4) in EOC with a major focus on cell–cell interactions. We examined the expression of CSPG4 in clinical samples and cell lines of EOC. The proliferation, migration, and invasion abilities of the CSPG4 knockdown cells were assessed. We also assessed the role of CSPG4 in spheroid formation and peritoneal metastasis in an in vivo model using sh-CSPG4 EOC cell lines. Of the clinical samples, 23 (44.2%) samples expressed CSPG4. CSPG4 was associated with a worse prognosis in patients with advanced EOC. Among the EOC cell lines, aggressive cell lines, including ES2, expressed CSPG4. When CSPG4 was knocked down using siRNA or shRNA, the cell proliferation, migration, and invasion abilities were significantly decreased compared to the control cells. Proteomic analyses showed changes in the expression of proteins related to the cell movement pathways. Spheroid formation was significantly inhibited when CSPG4 was inhibited. The number of nodules and the tumor burden of the omentum were significantly decreased in the sh-CSPG4 mouse models. In the peritoneal wash fluid from mice injected with sh-CSPG4 EOC cells, significantly fewer spheroids were present. Reduced CSPG4 expression was observed in lymphoid enhancer-binding factor 1-inhibited cells. CSPG4 is associated with aggressive features of EOC and poor prognosis. CSPG4 could be a new treatment target for blocking peritoneal metastasis by inhibiting spheroid formation.

## 1. Introduction

Epithelial ovarian cancer (EOC) is the most lethal gynecological malignancy, with an increasing number of patients. Because of a lack of reliable screening and specific symptoms at the early stage, more than 75% of EOC patients are diagnosed at advanced stages with peritoneal dissemination [1,2]. The treatment strategies for EOC include cytoreductive surgery and repetitive chemotherapy. However, nearly 80% of EOC patients at advanced stages exhibit the recurrence of peritoneal dissemination [3,4]. Therefore, the prognosis for patients with advanced EOC is poor. The 5-year survival rate is approximately 25% at advanced stages, and it has not been improved in recent decades [4,5]. In ascites, EOC cells form aggregated spheroids, which are reported to have a high metastatic ability in the peritoneal cavity [4,5]. However, the mechanisms underlying the spheroid formation and peritoneal dissemination of mesothelial cells, with respect to cell–cell interactions, are not fully understood [6].

Tumor-associated glycans have been reported to play a significant role in promoting the aggressive and metastatic behavior of malignant cells because of their important roles in cell–cell and cell–extracellular matrix interactions [7,8]. Among these glycans, chondroitin sulfate proteoglycan 4 (CSPG4), also known as neuron-glial antigen 2, is a surface type I transmembrane core proteoglycan and is crucially involved in cell survival, migration, carcinogenesis, and angiogenesis [6,9,10]. CSPG4 is often used as a marker for identifying and diagnosing several cell types or tumors [11,12,13,14]. And the expression level of CSPG4 is associated with poor prognosis in various tumors [15,16,17,18]. Moreover, CSPG4 is proposed as an attractive immunotherapy target because it has a restricted expression level in normal tissues [8].

Previous studies have revealed that inflammation and hypoxia, including transforming growth factor-beta1 (TGF-β) expression, are fundamental factors that increase CSPG4 expression [6]. Transcriptional factor paired box 3 (PAX3) also regulates CSPG4 expression upstream of the CSPG4 promoter region [6,14] However, the regulatory mechanisms involved in this process are not fully understood [6,15,16].

It is widely known that EOC cells can easily detach from the primary site in the form of spheroids and lead to peritoneal dissemination through their interaction with mesothelial cells. Therefore, we hypothesized that CSPG4 played an important role in EOC progression. However, the current evidence is limited regarding the role of CSPG4 in EOC [19]. The aim of this study was to reveal the functions of CSPG4 in EOC using clinical samples and CSPG4 knockdown EOC cell lines and whether CSPG4 could be a novel target for EOC treatment.

## 2. Results

### 2.1. The Positive Expression of CSPG4 in Clinical Samples and Cell Lines of EOC

Firstly, we determined whether CSPG4 was expressed in clinical samples and cell lines of EOC. We performed an immunohistochemical analysis of the CSPG4 expression in 52 EOC samples. Some samples showed strong staining in the cell membrane, as shown in Figure 1a. In total, 23 cases (44.2%) of EOC samples were positive for CSPG4 staining (Figure 1b). The CSPG4 expression was relatively weak in normal stroma and the epithelial surface of the ovaries compared to the ovarian cancer region (Appendix A). The mRNA expression level was significantly higher in ovarian cancer tissue than that in normal epithelial ovaries using data from a public database (Appendix A). Moreover, Western blot analysis revealed that six of the eight EOC samples were positive for CSPG4 (Figure 1c). Over 80% of EOC tissues expressed CSPG4 based on three kinds of antibodies for CSPG4 from the Human Protein Atlas database (n = 11–12) (Figure 1d), while its expression was not detected or was weak in normal ovarian samples (n = 3). We analyzed the Kaplan–Meier plotter database, including 1,081 patients with stage III or IV EOC with a follow-up period of 60 months. Patients with high CSPG4 expression had significantly shorter progression-free survival rates (HR = 1.23, P = 0.011) and worse overall survival (HR = 1.19, P = 0.029) (Figure 1e,f). The mRNA relative expression levels of CSPG4 in seven EOC cell lines were analyzed and compared with the positive control MDA-MB-231 and negative control MCF7 [16,18]. Among the EOC cell lines, HEY, ES2, A2780, and NOE were positive for CSPG4 expression (Figure 1g). The Western blot analysis also revealed the expression of CSPG4 in these cells (Figure 1h). Therefore, we used ES2 and NOE in the following knockdown experiments.

### 2.2. CSPG4 Downregulation Affects Invasion/Migration Abilities and Changes the Expression of Various Proteins in EOC Cells

We used small interfering RNAs (siRNA) to determine the role of CSPG4 in EOC. Western blot analysis and the mRNA relative expression level of CSPG4 with siRNA confirmed the knockdown efficacy of CSPG4 (Appendix A). Using a Transwell chamber system, we found that the migration and invasion abilities were significantly decreased when CSPG4 was knocked down (Figure 2a–c). Furthermore, we found that the wound-healing area was also significantly lesser with CSPG4-targeted siRNA than in the control cells (Figure 2d,e). The same experiments were performed using NOE cells, and the results were similar to those of ES2 (Appendix A). Moreover, the migration ability was decreased by using the antibody against CSPG4 (Figure 2f and Appendix A). 

Next, we examined the protein expression of ES2 cells using CSPG4-targeted siRNA and compared it to that of the control cells. The PCA map revealed that each group shared a similar protein expression and differed from each other (Figure 2g). The volcano plot revealed various changes in protein expression in the ES2 cells treated with the control or CSPG4-targeted siRNA (Figure 2h). The enriched ontology cluster showed significant changes in the cell movement pathways related to the actin cytoskeleton and muscle contraction (Figure 2i). These results support the importance of CSPG4 in EOC cells attaining the aggressive features of EOC.

### 2.3. EOC Cells with CSPG4 Knockdown Have Morphological Changes and Decreased Interactions with Mesothelial Cells

To evaluate the functions of CSPG4 in EOC cells in further experiments, including an in vivo model, we transduced ES2 cells with sh-CSPG4 or left them sh-negative. We confirmed a decreased CSPG4 expression on the cell surface and mRNA expression level (Appendix A). The EOC cells transfected with sh-CSPG4 were more rounded and had a lower cell aspect ratio (Figure 3a,b). The growth ability was significantly decreased in the ES2 cells treated with sh-CSPG4 compared to the control cells (Figure 3c). The EOC cells with sh-CSPG4 had low wound-healing abilities (Figure 3d,e and Appendix A). 

Our previous studies revealed that fibronectin 1 (FN1) was a key molecule interacting with EOC and mesothelial cells, especially malignant ascites [20]. In the mesothelial cells, FN1 was highly expressed, but CSPG4 was not (Figure 3f). The expression level of FN1 was significantly higher when the mesothelial cells were stimulated with TGF-β. The GFP-labeled sh-negative and sh-CSPG4 EOC cells were co-cultured with the mesothelial cells, and the movement of the EOC cells was traced (Figure 3g). The EOC cells with sh-CSPG4 showed minor movement on the surface of the mesothelial cells, and most of them appeared to stay at the same location. In contrast, the sh-negative EOC cells moved very quickly (Figure 3h). The movement of the EOC cells with sh-CSPG4 was significantly decreased in terms of the average movement speed per 20 min (Figure 3i) and the total movement distance for 10 h (Figure 3j) compared to the sh-negative EOC cells. During this time-lapse imaging, some sh-negative EOC cells were difficult to trace because their fluorescence signals suddenly disappeared or decreased, and then, after a while, their signals appeared and became strong again. Detailed observations revealed that the fluorescence levels of these cells decreased when they invaded the mesothelial cells. These incidents were not often observed in the EOC cells treated with sh-CSPG4. These results reveal that CSPG4 plays an important role in the interaction with mesothelial cells to induce peritoneal dissemination.

### 2.4. CSPG4 Downregulation Decreases the Ability to Form Spheroids

The ability of EOC cells to form spheroids is essential for them to survive in ascites and cause intraperitoneal metastasis. Therefore, spheroid formation was evaluated using an ultra-low attachment round-bottom plate with time-lapse imaging (Figure 4a). The areas of the spheres were marked using GFP (Figure 4b). As the spheres form, the average area of the spheres becomes smaller and more compact. The sh-negative ES-2 cells formed compact spheroids within 6 h of incubation, and their size did not change until 48 h (Figure 4c). In contrast, the sh-CSPG4 ES2 cells formed significantly fewer aggregated spheroids compared to the control cells from 3 h after the observation even though their area was not significantly different at the start point (Figure 4d,e). Moreover, the spheroids with sh-CSPG4 were loose and increased in area after 24 h, indicating that these spheroids were not aggregated (Figure 4d). These results show the importance of CSPG4 in the spheroid formation of EOC cells in ascites. 

### 2.5. CSPG4 Is Related to Peritoneal Dissemination and Sphere Formation in Ascites of Mice

To assess the capacity of EOC cells for peritoneal dissemination, we injected GFP-labeled sh-CSPG4 or sh-negative EOC cells into the peritoneal cavity in mice (Figure 5a). The mice with sh-CSPG4 ES2 cells exhibited fewer numbers of peritoneal metastatic nodules than those with sh-negative ES2 cells (Figure 5b,c). In addition, the weight of the omentum was significantly higher in the mice with sh-negative ES2 cells than in the sh-CSPG4 mice (Figure 5d). We also found many aggregated spheroids in the peritoneal wash fluid of mice injected with sh-negative ES2 cells. In contrast, the number of spheroids was significantly lower in the fluid from the mice with sh-CSPG4 (Figure 5b,e). IVIS imaging revealed that the EOC cells from the sh-CSPG4 mice had inhibited growth in the right ovary (Figure 5f). Moreover, the pathological findings in the omentum revealed that the ES2 tumors with sh-CSPG4 showed less aggressive morphological features with a smaller nucleus size and less frequent mitotic figures (Figure 5g). The tumors with sh-CSPG4 showed a decreased proliferation ability, as well as a significantly decreased expression of the proliferation marker Ki67 (Figure 5g,h). These results suggest the importance of CSPG4 in peritoneal metastasis in a mouse model.

### 2.6. Lymphoid Enhancer-Binding Factor 1 (LEF1) Regulates CSPG4 Expression in EOC

Previous reports have shown that CSPG4 is regulated by PAX3 [6,14]. A qPCR analysis of PAX3 expression was performed in various EOC cell lines with high CSPG4 expression (Appendix A). *PAX3*-targeted siRNA decreased the PAX3 expression without affecting the expression of CSPG4 (Figure 6a). To identify the regulator of CSPG4 in EOC, we fully checked upstream of the *CSPG4* and found that many LEF1-binding sites existed (Figure 6b). Therefore, we examined the effect of LEF1 siRNA on CSPG4 expression. The relative mRNA expression level of both LEF1 and CSPG4 was significantly decreased in cells treated with LEF1-targeted siRNA (Figure 6c). Moreover, flow cytometry analysis revealed that the surface expression of CSPG4 significantly decreased in the EOC cells treated with *LEF1*-targeted siRNA (Figure 6d). Similar to CSPG4, LEF1 is not regulated by PAX3 (Appendix A). Moreover, we used HeLa cells as a simple method to reveal the relationship between LEF1 and the promoter of *CSPG4,* and we revealed that the *LEF1*-induced HeLa cells had a strong expression of CSPG4 compared to the control cells (Figure 6e), which meant LEF1 affected the promoter area of CSPG4. Finally, to reveal the relationship in the EOC cells, we transfected pGL-4.1-*CSPG4*-luc or pGL-4.1-mock-luc (as a control) after control/*LEF1* siRNA induction in the EOC cells. As Figure 6f shows, when LEF1 was inhibited, the fluorescence level was significantly lower compared to that of the control in the ES2 cells. The fluorescence level was low in OVCAR3 regardless of the transfection because OVCAR3 does not have endogenous LEF1 expression. This indicated that the luciferase activity of the promoter of *CSPG4* was dependent on the expression of LEF1. These results reveal that LEF1, but not PAX3, could be a key regulator of the CSPG4 expression in EOC cells.

## 3. Discussion

In this study, we revealed that CSPG4 played an important role in peritoneal dissemination through the migration/invasion ability and cell–cell interactions in EOC. We confirmed that CSPG4 was expressed in EOC clinical samples and associated with the proliferation and migration of the EOC cells. CSPG4 downregulation remarkably decreased the ability of the EOC cells to form spheroids in both 3D and in vivo models. Further, we found that sh-CSPG4 treatment significantly decreased peritoneal dissemination and tumorigenesis. Moreover, our results suggested that LEF1 is essential in regulating CSPG4; however, the expression of PAX3, which has been reported to be a regulator of CSPG4, was not associated with the CSPG4 expression in the EOC cells.

It is well known that CSPG4 plays an important role in cell adhesion, motility, and invasion in various types of malignant tumors [11,15]. However, the evidence regarding the role of CSPG4 in EOC is limited [19]. In this study, we revealed that CSPG4 was expressed in nearly half of the clinical EOC samples and related to a poor prognosis at stage III or IV. CSPG4 knockdown in the EOC cells changed the cell’s morphology to a round shape. Some studies have shown that CSPG4 expression in malignant melanoma is related to the morphological changes associated with the epithelial-to-mesenchymal transition (EMT) [21]. Similarly, previous studies have revealed that cell lines with CSPG4 expression exhibit mesenchymal markers and high invasion properties [22,23]. These morphological changes may affect the migration and invasion abilities of cancer cells. Actually, the downregulation of CSPG4 decreased the migration, invasion, and wound-healing abilities of the EOC cell lines. We also revealed the changes in the expression of proteins, especially those involved in the cell movement pathways related to the actin cytoskeleton and muscle contraction, in EOC cells with CSPG4-targeted siRNA. Therefore, CSPG4 plays an important role in the aggressive features of EOC.

EOC has a unique metastatic root that uses ascites [4]. To induce peritoneal metastasis, EOC cells need to invade the mesothelial layer, which consists of mesothelial cells. Therefore, the interaction between EOC and the mesothelial cells is critical for the formation of peritoneal metastasis [23]. It is at this point that the primary mesothelial cells express FN1, which binds to the extracellular domain of CSPG4 [24]. Previous studies have revealed that the TGF-β secreted by EOC cells in malignant ascites alters the conditions of the mesothelial cells [20,25]. Interestingly, the FN1 expression was significantly higher when the mesothelial cells were stimulated with TGF-β. Therefore, TGF-β secretion further promoted the interaction between EOC and the mesothelial cells. Moreover, we found that EOC cells with CSPG4 knockdown had restricted movement on the surface of the mesothelial cells. These movements can aid EOC cells to find the proper metastatic area, including the milky spots of the omentum [26]. Although we could not assess the interaction between the mesothelial cells and EOC spheroids, these results highlight that blocking CSPG4 in the EOC cells decreased their interaction with the mesothelial cells, leading to decreased peritoneal metastasis.

Accumulating ascites is one of the unique characteristics of EOC. In ascites, EOC cells do not exist as a single cell but form aggregated spheroids. Assessing their ability to form spheroids is essential to understanding the metastatic ability of EOC. Previous studies have revealed that the expression of CSPG4 in human melanoma cells promoted their resistance to anoikis [27,28]. Interestingly, we found that CSPG4 knockdown remarkably decreased the spheroid formation of EOC cells. In addition, our in vivo model study showed that sh-CSPG4 transfection significantly decreased the spheroid formation of ES2 cells in peritoneal wash fluid. Although a previous study showed that CSPG4 was related to spheroid formation [19], our model is much closer to the clinical situation. Together, CSPG4 strongly affects the formation of spheroids in EOC cells, and these characteristics can strengthen the aggressiveness of EOC in surviving and forming peritoneal metastases in ascites.

Although many studies have investigated the function of CSPG4 in various tumors, little is known regarding how CSPG4 expression is regulated [15]. Previous studies have revealed that PAX3 expression is related to CSPG4 expression [6,14]. Although PAX3-targeted siRNA decreased the expression of PAX3, the expression of CSPG4 did not change. From the upstream sequence of the CSPG4 coding area, we found that many LEF1-binding motifs existed. Indeed, LEF1 was found to regulate the CSPG4 promoter. In colorectal cancer, LEF1 is well known to have critical role in cancer stem-like cell survival and self-renewal [29], and disruption of the LEF1 pathway can be reduce recurrence [30]. In addition, a previous study revealed that LEF1 was consistently expressed in the serous tubal intraepithelial carcinoma, the precursor region of ovarian cancer [31]. To our knowledge, this is the first report on the relationship between CSPG4 expression and LEF1 regulation. CSPG4 maintains extracellular signal-regulated kinase 1/2 (ERK1/2) activation through receptor tyrosine kinase (RTK) signaling via the mitogen-activated protein kinase (MAPK) cascade. Meanwhile, focal adhesion kinase (FAK) activation is maintained through integrin signaling, and FAK plays an important role in tumor growth and invasion [32,33]. FAK also plays an important role in the regulation of cell migration, invasion, adhesion, proliferation, and viability in ovarian cancer [34,35], and FN1 is involved in the upregulation of the FAK pathway to stimulate ovarian cancer cell migration and invasion [36]. In addition, the involvement of CSPG4 in the adhesion and motility of melanoma cells on FN1 substrates has been reported [27,28]. Therefore, LEF1 could be an important transcriptional factor related to the aggressive features of EOC, while the expression of CSPG4 and CSPG4 in ovarian cancer cells is predicted to activate the FAK pathway through its interaction with FN1 and promote the migration and invasion of ovarian cancer cells via downstream signaling (PI3K/Akt).

CSPG4 is considered a suitable target for immune-based therapies because of its restricted expression in a few cell types in adults [8,12,37,38]. Moreover, previous reports have shown no obvious deleterious side effects when using immunotherapy against CSPG4 [15]. Many studies have revealed that anti-CSPG4 monoclonal antibodies or lentiviruses encoding a CSPG4-targeted shRNA in tumors restricted tumor growth and metastasis [39,40]. In this study, the deletion of CSPG4 reduced the spheroid formation and tumor burden in peritoneal metastasis. These abilities may be related to cell–cell interactions. Therefore, CSPG4 may be a suitable target for blocking peritoneal metastasis and drug resistance in EOC patients.

This study has some limitations. First, we did not reveal the molecular mechanisms underlying CSPG4 expression and spheroid formation. Moreover, some studies have reported that CSPG4 is expressed in the endothelial cells and affects vascular formation in the tumor microenvironment [8]. In this study, we did not examine the construction of blood vessels in the tumors. Analyses based on patient demographics and subtypes related to CSPG4 expression were not performed. CPSG4 was related only to patients at the advanced stages in the specific dataset. Although we performed several experiments with two different cell lines which expressed mRNA of CSPG4, we did not focus on the histology of EOC. The ES2 cell line represented low-grade serous carcinoma in genetic expression, while ES2’s growth is often fast [40,41]. We did not assess the differences in histology in EOC. Further studies are needed to address what role CSPG4 plays exactly in EOC and its impact on EOC outcomes.

In conclusion, this study showed that CSPG4 was expressed in EOC and related to the tumor aggressiveness, including proliferation, invasion, and sphere formation. Specifically, we revealed that CSPG4 was related to spheroid formation in EOC in both 3D and in vivo models. Downregulation of CSPG4 was significantly associated with decreased tumor burden and peritoneal metastasis. This ability may be related to the development of peritoneal metastasis. In EOC, CSPG4 was regulated by LEF1 but not by PAX3. CSPG4 has restricted expression in normal tissues and the ability to control peritoneal dissemination; therefore, CSPG4 could be a suitable treatment target for EOC by controlling peritoneal metastasis, which is the most important strategy to improve the prognosis in EOC patients.

## 4. Materials and Methods

### 4.1. Patient Sample Collection

The clinical samples were collected from patients diagnosed with EOC who had undergone surgery at Nagoya University Hospital. From January 2013 to December 2018, we collected ovarian cancer tissues which had primary tumors and metastasis. We detected 52 patients and used those samples for immunohistochemistry (Appendix A). Most of them were diagnosed with high-grade serous carcinoma, except for some patients with clear cell carcinoma. Among them, 8 cases of high-grade serous carcinoma were selected, and we performed Western blot analysis to check the CSPG4 expression. Informed consent was obtained from all patients according to the regulations set out by the Ethics Committee of Nagoya University (Approval number: 2017–0497; approved date: 13 March 2018).

### 4.2. Immunohistochemistry

Immunohistochemistry of the samples was performed using an anti-human CSPG4-specific mouse monoclonal antibody (D2.8.5-C4B8, created by Dr. Ferrone) 1:300 according to previous reports and the manufacturer’s instructions [10,42]. Formalin-fixed paraffin-embedded ovarian cancer tissues were cut into 4 µm thick sections, dried, deparaffinized, and rehydrated with water using graded alcohols. Endogenous peroxidase was blocked with 1% H_2_O_2_ (95321–100ML, Sigma-Aldrich, St. Louis, MO, USA). Heat-Induced Epitope Retrieval was carried out by using target retrieval solution at a pH of 6.0 (S1699, DAKO, St. Clara, CA, USA) + 0.2% Triton-X100 (T8787–50ML, Sigma, Tokyo, Japan) for 20 min at 95 °C. Thereafter, these slides were incubated with DAB (K3477, DAKO) for 5 min and counterstained with Mayer’s hematoxylin (Wako, Osaka, Japan) for 30 sec. The intensity of CSPG4 immunostaining was scored as follows: 0 (negative), 1 (moderate), and 2 (strong). Tumors with a score of 1 or 2 were defined as positive for CSPG4 expression [10]. To analyze the proliferation ability, staining with the anti-Ki67 antibody was performed (ab9260, Abcam, Cambridge, UK) 1:150. The Ki67 expression rate in the tumor tissue was calculated in three randomly selected fields in each sample. Negative controls were run on all sections in blocking buffer generated against unrelated antigens.

### 4.3. Western Blot Analysis

Western blot analysis was performed to detect the CSPG4 antigens in clinical samples from fresh biopsy samples and cell lines. The clinical samples and cells were lysed in sample buffer (62.5 mM Tris-HCl [pH 8.0], 2% sodium dodecyl sulfate [SDS], 10% glycerol, 0.002% bromophenol blue, and 100mM dithiothreitol [DTT]). These samples were separated using SDS polyacrylamide gel electrophoresis and blotted onto an Immobilon-P membrane (Merck Millipore, Burlington, MA, USA). Immunoreactive signals were detected via enhanced chemiluminescence (GE Healthcare Systems, Chicago, IL, USA) using the ImageQuant LAS 4000 mini kit (GE Healthcare Systems, Singapore). The antibodies (Abs) utilized were anti-CSPG4 (763.74) and anti-GAPDH (CST, Tokyo, Japan).

### 4.4. Public Data Analysis

The associations between the prognoses of EOC patients were analyzed using a microarray gene expression database with a Kaplan–Meier plotter depending on the mRNA expression of CSPG4 (http://kmplot.com/analysis/index.php?p=service&cancer=ovar) (accessed on 10 April 2022). We used 214297_at as the gene symbol for CSPG4 and included patients at stages 3–4 with a follow-up threshold of 60 months. The protein expression of CSPG4 was analyzed using THE HUMAN PROTEIN ATLAS (https://www.proteinatlas.org/) (accessed on 16 August 2022). The difference in the mRNA expression of CSPG4 in normal ovarian surface and ovarian cancer cells was analyzed at CSIOVDB (http://csiovdb.mc.ntu.edu.tw/CSIOVDB.html accessed on 21 October 2023), which includes 3431 data points on human ovarian cancer.

### 4.5. Quantitative Real-Time PCR (qPCR)

We used a semi-quantitative PCR method to detect the mRNAs in the human ovarian cancer cell lines. The total RNA was extracted from the cell lines using the RNeasy Mini Kit (QIAGEN, Tokyo, Japan) and subjected to qPCR. The cDNA was synthesized from 1 µg of total RNA using a High-Capacity cDNA Reverse Transcription Kit (Applied Biosystems, Tokyo, Japan) with an oligo dT primer. Fast SYBER Green Master Mix (Applied Biosystems) was used for amplification and the samples were run on the StepOne system (Applied Biosystems). The primers were designed to detect spliced mRNA, and their sequences are shown in Appendix A. All the primers were synthesized by and purchased from Hokkaido System Science.

### 4.6. Cell Lines

We used several established EOC cell lines, ES2, OVCAR-3, OV-90, SK-OV-3, A2780, HEY, and NOE, to detect the CSPG4 expression. The ES2, OVCAR-3, OV-90, SK-OV-3, and HeLa cells were obtained from the American Type Culture Collection (Manassas, VA, USA). The A2780 cells were purchased from the European Collection of Authenticated Cell Cultures (Salisbury, UK). The HEY cells were purchased from Cosmo Bio (Tokyo, Japan). The breast adenocarcinoma cell lines MDA-MB-231 and MCF7 were from Sumitomo Pharma (Osaka, Japan). NOE was established in our department from the primary EOC, as described previously [43]. The breast adenocarcinoma cell lines MDA-MB-231 and MCF7 were used as the positive and negative controls for CSPG4 expression, respectively [16,44]. We used these cell lines because base-like breast cancer is related to BRCA1 mutation, which is also an important mutation in EOC [18]. To generate ES2 and NOE that stably expressed green fluorescent protein (GFP) or luciferase, a recombinant retrovirus was used, as described previously [22,23]. The human primary mesothelial cells were obtained as described previously [20,23]. Briefly, the human primary mesothelial cells were obtained from the omentum without apparent tumor dissemination. The omentum resected during the operation was cut into 4 cm^2^ pieces and trypsinized at 37 °C for 30 min with continuous shaking. The contents were passed through a 200 μm mesh filter and then centrifuged at 300× *g* for 5 min. The collected cells were cultured on collagen-coated dishes.

### 4.7. Knocked-Down CSPG4 Expression

To inhibit CSPG4 expression, we designed two siRNAs against CSPG4 (#1 and #2) as described before [22]. A non-targeting siRNA was used as a negative control. All siRNAs were synthesized by Hokkaido System Science (Sapporo, Japan). The siRNAs were transfected into the EOC cells for 48 h. pSIREN-RetroQ-DsRed-Express vectors encoding small hairpin RNA (shRNA) against CSPG4 (sh-CSPG4) and a non-target (sh-negative) were used to inhibit the CSPG4 expression. Both vectors were purchased from Clontech (Takara Bio, Kusatsu, Japan). The detailed sequences of each siRNA and shRNA are summarized in Appendix A. Their effectiveness was confirmed via Western blot analysis and verified using qPCR. The total RNA isolated from the siRNA-transfected cells or shRNA-transduced cells was used to synthesize the cDNA. GAPDH was used for normalization.

### 4.8. Transwell Migration and Invasion Assays

The migration assay was performed in a Transwell chamber (8 μm, Corning Japan, Tokyo, Japan). The invasion assay used a BioCoat Matrigel Invasion Chamber (8 μm, Corning). The cells were treated with the desired siRNA or shRNA. Briefly, the cell suspensions (5 × 10^4^ cells/mL) were placed on top of the upper chamber (500 μL/well). During the 22 h incubation period, the cells moved through the membrane or matrix and adhered to the bottom membrane of the insert. Motile cells were fixed with methanol and then stained with May–Grünwald Giemsa stain. Images were obtained using an BX43 upright microscope (Olympus, Tokyo, Japan) and analyzed using the ImageJ software (version 1.53e) (NIH, Bethesda, MD, USA).

### 4.9. Wound Healing

The wound-healing assay used a Culture-Insert 2 Well (ibidi, Nippon Genetics, Tokyo, Japan). The cells were treated with the desired siRNA. Briefly, a cell suspension (7 × 10^5^ cells/mL) was applied to each well (70 μL/well) and then placed in an incubator overnight. The Culture-Insert well was removed gently, and the dish was filled with cell-free medium. Images of the wounds were obtained using an inverted CKX41 microscope (Olympus, Tokyo, Japan), and the wound healing was quantified using the ImageJ. The percentage of the original wound area that had healed was defined as follows: [(area of original wound−area at subsequent time point) / area of original wound] × 100 (%).

### 4.10. Proteomic Analysis

The following software and packages were used: R (V4.1.1), RStudio (2021.09.0), the tidyverse package (V1.3.1), the missForest package (1.5), the limma package (V3.48.3), the mixOmics package (V6.16.3), and the clusterProfiler package (V4.0.5). Missing values were imputed using the MissForest package. All the values in the expression matrix were log-transformed and median-centered. Differential abundance analysis was conducted using the limma package, and principal component analysis (PCA) was carried out using the mixOmics package.

### 4.11. Immunofluorescence

The cells were incubated on glass coverslips. After demonstrating appropriate attachment, the cells were fixed using 4% PFA in PBS. They were then washed with PBS and treated with phalloidin-594 and DAPI at room temperature for 1 h. After thorough washing, the cells on the coverslips were overlaid with a mounting medium (Cosmo Bio, Tokyo, Japan) and then analyzed using an FV1000 (Olympus, Tokyo, Japan).

### 4.12. Sphere Forming Assay

The GFP-labeled EOC cells (2000 cells/well) were plated in 96-well round-bottom ultra-low attachment microplates (Corning 7007, New York, NY, USA). Immediately after plating, we continuously checked for sphere formation every 1 h for 48 h using an IncuCyte ZOOM (Essen BioScience Inc., Ann Arbor, MI, USA) with a bright field and green fluorescence. The spheroid area was analyzed using green fluorescence light (hole fill: μm^2^; minimum filter: 100 μm^2^) using Zoom Base Analysis (Essen BioScience). The fluorescent images were detected according to the top-hat background subtraction of each image (radius: 100 μm, threshold: 2.0 GCU).

### 4.13. Mesothelial Migration Assay

The mesothelial cells were cultured on 6-well collagen-coated plates in RPMI-1640 media supplemented with 10% FBS and penicillin/streptomycin until confluency. On a monolayer of mesothelial cells, 5 × 10^4^ GFP-labeled ES-2 single cells transfected with control or shCSPG4 were seeded. The time-lapse images were acquired every 10 min for 10 h using a BZX-800 microscope (KEYENCE, Osaka, Japan) and then analyzed using BZX-800 View Analyzer software (KEYENCE). The average migration velocities and mean squared displacements (MSDs) of individual cancer cells were obtained.

### 4.14. Mouse Model Studies

To establish a model of malignant peritonitis, 1 × 10^6^ GFP-labeled ES2 cells induced as sh-negative or with sh-CSPG4 were injected into the peritoneal cavity of 6- to 8-week-old female BALB/c nude mice. The mice were maintained under specific pathogen-free conditions. One week after injection, the mice were sacrificed. Before laparotomy, 1 mL of PBS was injected using a 22 Gy needle and washed for 30 s. Then, the fluid was collected using an 18 Gy needle as peritoneal wash fluid. From the collected fluid, 300 μL was centrifuged for 5 min. After removing the supernatant, the cells were gently pipetted with 100 μL of 4% PFA and observed in terms of the spheroids in the fluid. The abdominal cavity was observed, and the number of peritoneal disseminations was counted. The omentum was removed, and the weight was measured. The peritoneal cavity and resected omentum were observed under an M205FA fluorescence microscope (Leica, Wetzlar, Germany). The resected omentum was fixed with 4% PFA overnight and used for further immunohistochemistry experiments. ES2 luc-sh-negative or sh-CSPG4 cells were inoculated into the ovaries of the nude mice. This orthotopic inoculation was described previously [22]. Briefly, female nude mice at 8 weeks of age were used. The cells (1 × 10^5^ cells/μL in medium/Matrigel (1:1) solution) were injected into the murine ovaries (1 μL/ovary/mouse) using a retroperitoneal approach from the dorsal flank. After inoculation, the tumor growth was visualized via intraperitoneal injection with luciferin (1.5 mg/ 10 g body weight, Promega, Tokyo, Japan) and analyzed using the IVIS imaging system (PerkinElmer, Waltham, MA, USA).

### 4.15. Analysis of Cell Surface Markers

The expression of CSPG4 on the cell surface was examined using flow cytometry. Briefly, 106 cells were stained with Alexa Fluor 647-conjugated mouse anti-CSPG4 monoclonal antibody, LHM2 (Santa Cruz Biotechnology, Dallas, TX, USA), for 30 min on ice, washed three times with 1% BSA in PBS, and then suspended with 1% BSA in PBS prior to analysis using a FACSAria II (BD Biosciences, Tokyo, Japan).

### 4.16. Promoter Luciferase Assays

We used HeLa cells because this cell line demonstrated a high transfection efficiency for many genes, allowing for the effective introduction of foreign genes. The HeLa cells were transfected with CSPG4 Firefly Enhancer luciferase (luc) inserted into pGL4.10 (Promega) and the HSV thymidine kinase promoter (pRL-TK) with a Renilla luc expression vector (Promega). AcGFP vectors with lymphoid enhancer-binding factor 1 (LEF1) or the control were also injected (Clontech), as described before [22]. Then, 48 hours after transfection, the luciferase activity was assayed using the Dual-Luciferase Reporter Assay kit (Promega) and a Turner Veritas Microplate Luminometer or a BioTek Synergy HTX Multimode Reader. The relative luciferase activity for each construct was obtained by dividing the Firefly luciferase activity by the Renilla luciferase activity.

### 4.17. Statistical Analysis

To compare the differences between two groups, a two-tailed Student’s t-test or Mann–Whitney U test was used. A significant difference was set at *p* < 0.05. All statistical analyses were conducted using IBM SPSS Statistics version 27.0 (IBM Corp., Armonk, NY, USA).

## Figures and Tables

**Figure 1 ijms-25-01626-f001:**
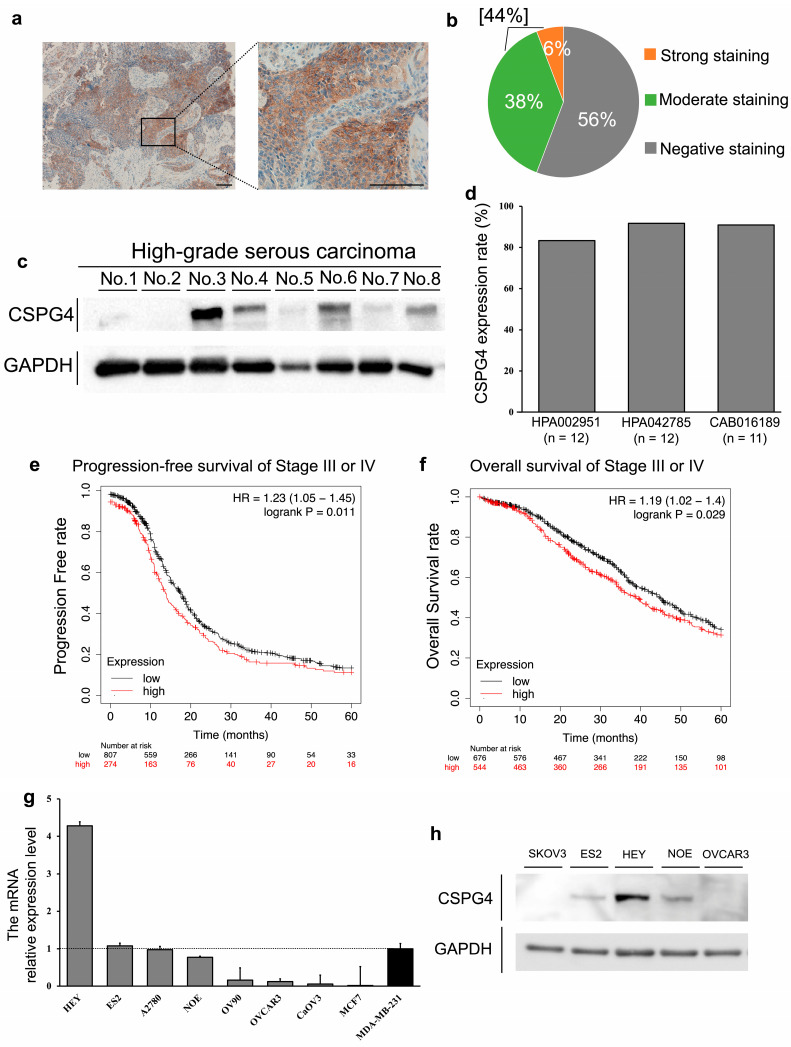
CSPG4 expression in clinical samples and various cell lines of EOC. (**a**) Representative image of strongly expressed CSPG4 with staining in clinical EOC samples. CSPG4 is strongly expressed in the cell membranes of the tumor cells. The scale bar is 100 μm. (**b**) Of 52 EOC samples, 23 (44.2%) cases showed positive CSPG4 expression. (**c**) Western blot analyses with the anti-CSPG4 antibody. (**d**) Three kinds of antibodies against CSPG4 showed over 80% CSPG4 expression in EOC tissues for each antibody. (**e**) The Kaplan–Meier analysis included 1,081 patients with EOC at stage III or IV. Progression-free survival rate was significantly shorter in the patients with high expression of CSPG4. (**f**) Overall survival was significantly lower in EOC patients with high expression of CSPG4. (**g**) mRNA levels of each EOC cell line measured via RT-qPCR. HEY, ES2, A2780, and NOE cells had high mRNA levels of CSPG4 compared to the positive control MDA-MB-231. (**h**) Western blotting analyses of CSPG4 in EOC cell lines.

**Figure 2 ijms-25-01626-f002:**
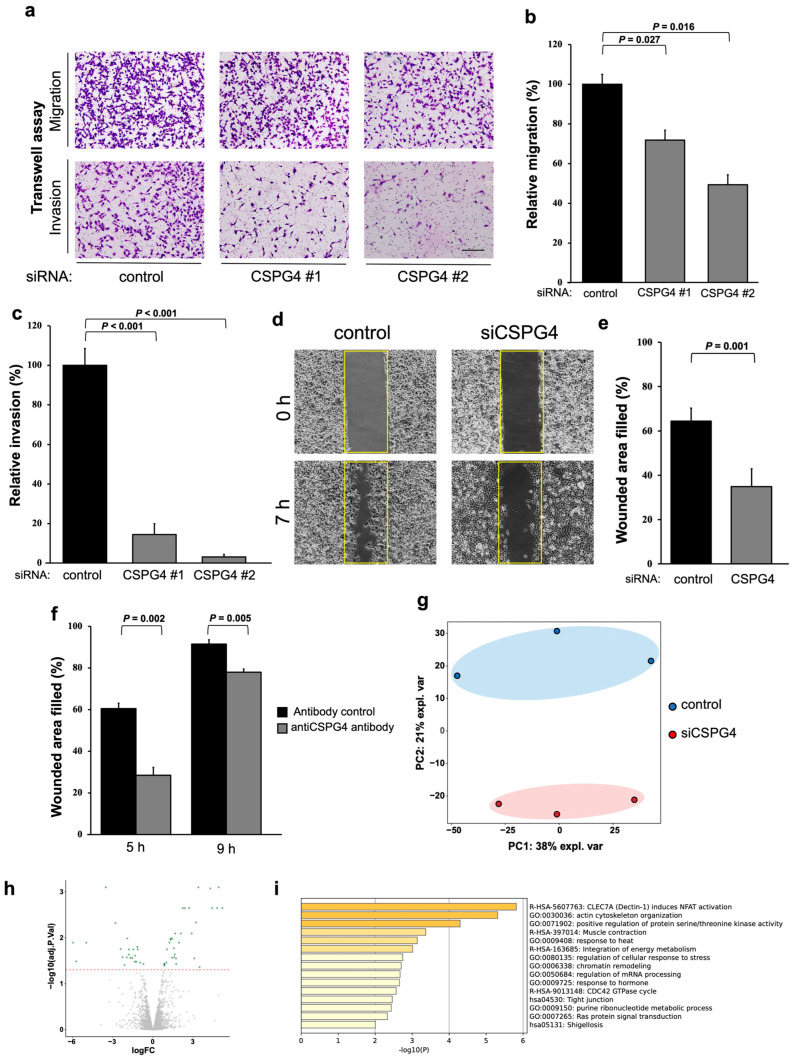
Downregulation of CSPG4 affects invasion/migration abilities and changes various protein expressions in EOC cells. (**a**), Representative images of decreased invasion/migration assessed using Transwell assay using CSPG4-targeted siRNA. The scale bar is 50 μm. (**b**,**c**) Both CSPG4-targeted siRNA#1 and #2 significantly decreased migration/invasion abilities. (**d**) Representative images of wound-healing assay using EOC cells with the control or CSPG4-targeted siRNA. Yellow area represents of the first scratch. (**e**) ES2 cells transfected with CSPG4-targeted siRNA had smaller wound-healing area compared to control cells. (**f**) Wound-healing ability was decreased when EOC cells were treated with a CSPG4 antibody. (**g**) Principal component analysis (PCA) plots of proteomic analyses of ES2 cells with and without CSPG4-targeted siRNA. (**h**) Volcano plot analyses revealed that various proteins were upregulated or downregulated by CSPG4 downregulation. (**i**) Pathway analyses revealed that cell movement pathways related to the actin cytoskeleton and muscle contraction were affected by CSPG4-targeted siRNA.

**Figure 3 ijms-25-01626-f003:**
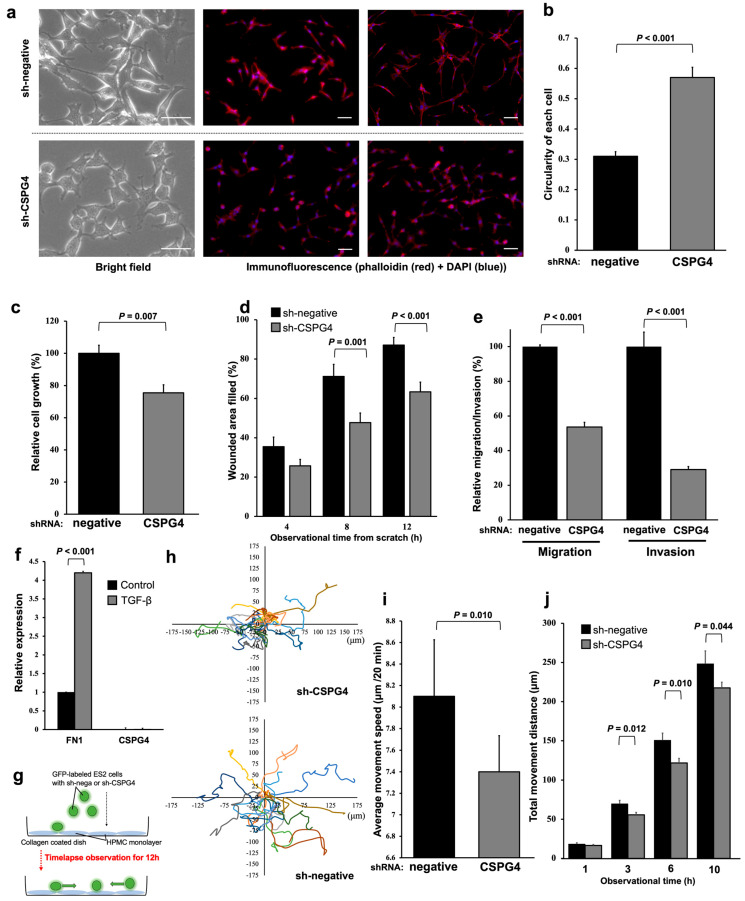
EOC cells with CSPG4 knockdown had morphological changes and decreased interactions with mesothelial cells. (**a**) ES2 cells were rounder with a lower cell aspect ratio when transfected with sh-CSPG4. The scale bar is 50 μm. (**b**) ES2 cells with sh-CSPG4 had higher circularity compared to control cells. (**c**) sh-CSPG4 cells had a significantly decreased growth rate than control cells. (**d**) Compared to sh-control cells, the wound-healing ability was significantly suppressed in ES2 cells with sh-CSPG4. (**e**) Migration and invasion abilities assessed using the Transwell incubation system significantly decreased in ES2 cells with sh-CSPG4. (**f**) Fibronectin1 (FN1) and CSPG4 expression in mesothelial cells. Mesothelial cells expressed FN1, and the expression significantly increased when mesothelial cells were stimulated with TGF-β. (**g**) Images of interaction and movement of EOC cells on a single layer of mesothelial cells. (**h**) ES2 cells with sh-CSPG4 and control cells showed different movements on the mesothelial layer. Each analyze cell shows different color. (**i**) ES2 cells with sh-CSPG4 had significantly less movement in an average of 20 min. (**j**) Based on the total movement distance, ES2 cells with sh-CSPG4 showed significant suppression in movement at all time points after 3 h.

**Figure 4 ijms-25-01626-f004:**
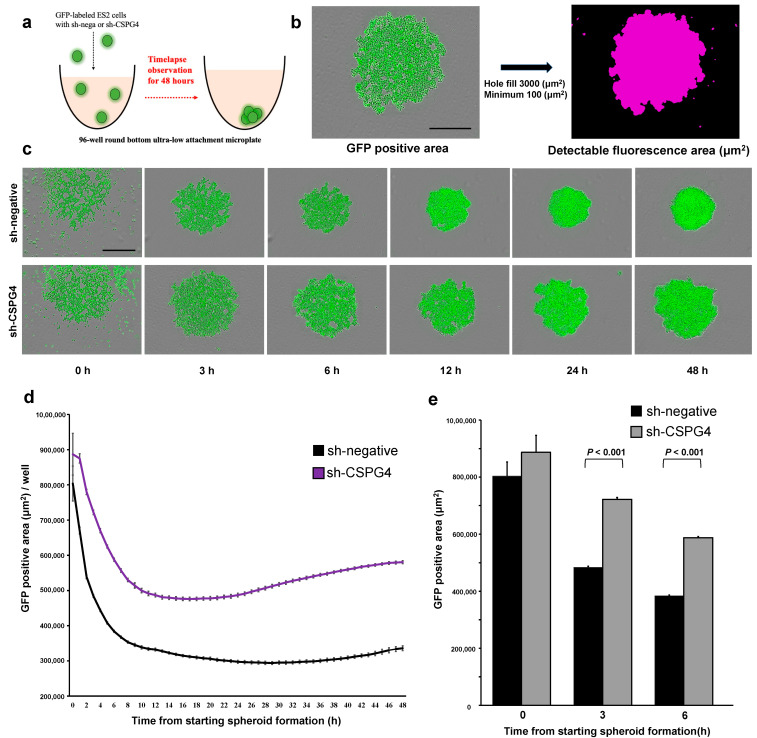
CSPG4 downregulation decreased spheroid formation in EOC cells. (**a**) Schematic protocol used for sphere formation assay. (**b**) Representative images for calculating spheroid area in each well. The evaluation was performed using GFP signals. The scale bar is 100 μm. (**c**) Representative images of spheroid formation at 0, 3, 6, 12, 24, and 48 h in sh-CSPG4 or sh-negative ES2 cells. Control cells formed compact spheroids rapidly. The scale bar is 100 μm. (**d**) The changes in the spheroid GFP-positive area in each group every 1 h for 48 h. ES2 cells with sh-CSPG4 formed less aggregated spheroids and the GFP area was significantly wider than that of sh-negative cells. (**e**) ES2 cells with sh-CSPG4 did not form compact spheroids compared to control cells.

**Figure 5 ijms-25-01626-f005:**
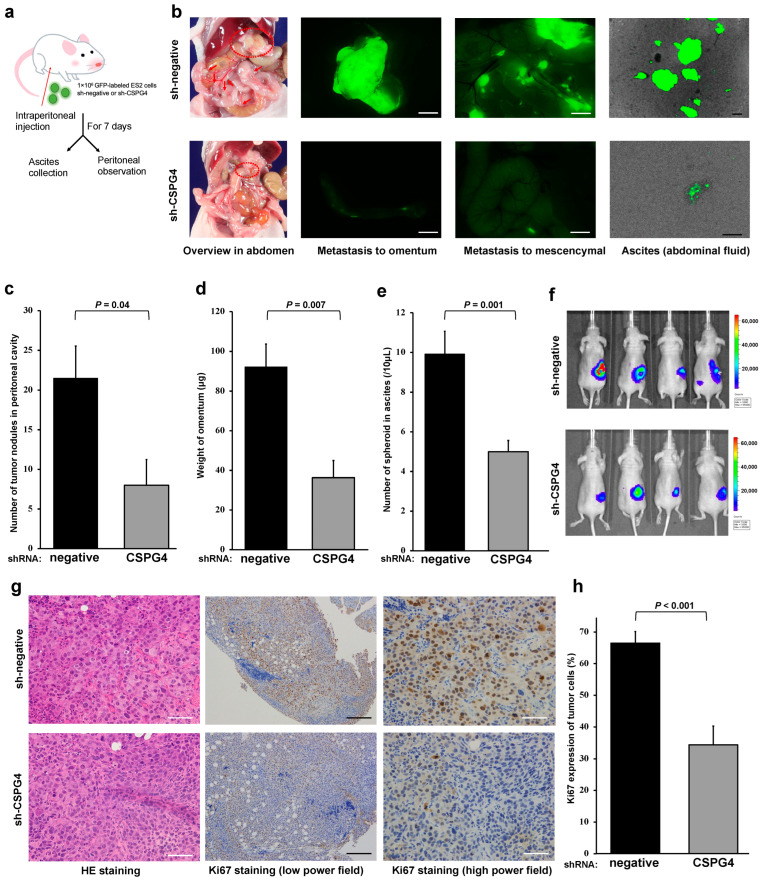
CSPG4 is related to peritoneal dissemination and sphere formation in ascites of mice. (**a**) Schematic protocol used for a malignant peritonitis model with GFP-labeled ES2 cells. (**b**) Representative images of peritoneal dissemination and spheroid formation in the peritoneal wash fluid. Red circle represents metastasis of omentum and red arrows represent peritoneal metastases. The white scale bar is 2.5 mm. The black scale bar is 200 μm. (**c**) The number of peritoneal disseminations was significantly decreased for ES2 cells with sh-CSPG4 (n = 4 in each group). (**d**) The weight of the omentum in the control group was significantly higher than that of sh-CSPG4 mice. (**e**) The number of spheroids was significantly less in the peritoneal wash fluid from the mice with sh-CSPG4 ES2 cells compared to control mice. (**f**) Representative model of orthotopic inoculation images. (**g**) Representative images of pathological findings of the omentum in each group. Tumors with sh-CSPG4 showed less aggressive morphological features and lower Ki67 (a proliferation marker) expression. The white scale bar is 50 μm. The black scale bar is 200 μm. (**h**) Ki67 expression was significantly decreased in the tumors with sh-CSPG4 compared to sh-negative tumors.

**Figure 6 ijms-25-01626-f006:**
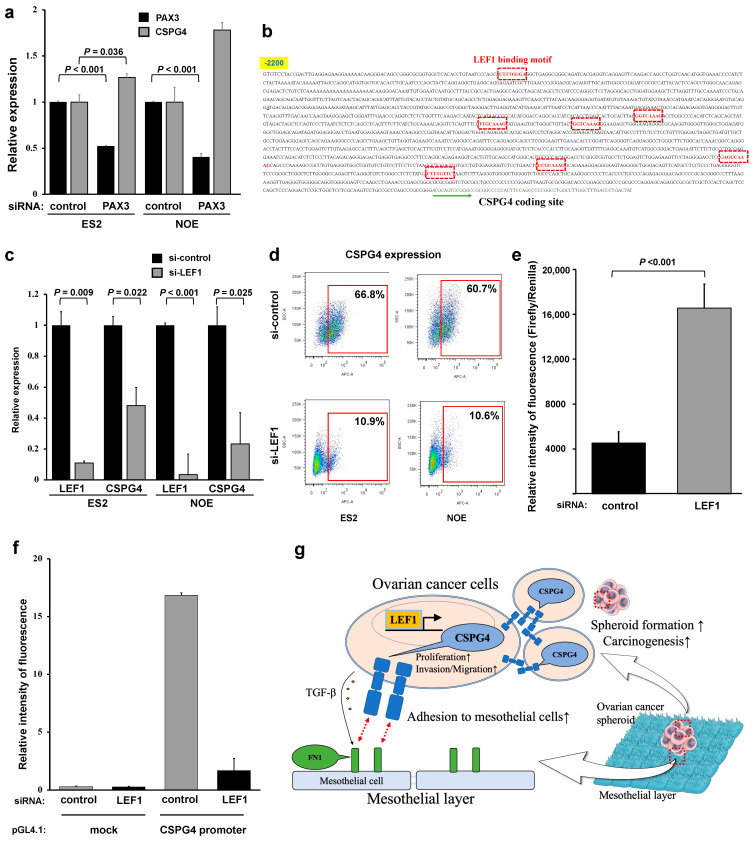
LEF1 regulates CSPG4 expression in EOC, not PAX3. (**a**) mRNA levels of PAX3 and CSPG4 in cells treated with PAX3-targeted siRNA. CSPG4 expression was not affected by PAX3-targeted siRNA. (**b**) Genetic information of the CSPG4 coding site and the region upstream of CSPG4. Several binding motifs of LEF1 (red square) were found. (**c**) LEF1 and CSPG4 expression in ES2 and NOE cells with LEF1-targeted siRNA. Not only LEF1 but also CSPG4 expression was affected by LEF1-targeted siRNA. (**d**) Flow cytometry analysis of the surface expression in cells with LEF1-targeted siRNA. (**e**) LEF1 regulated the expression of CSPG4 to affect the promoter site of CSPG4. (**f**) When LEF1 was inhibited with siLEF1, the fluorescence level was significantly lower compared to that of the control in ES2 cells. (**g**) Graphic abstract of this study. CSPG4 plays an important role in proliferation, invasion/migration, and forming spheroid formation in EOC. EOC cells expressing CSPG4 have strong interactions with mesothelial cells through FN1. LEF1 regulates CSPG4 expression in EOC cells.

## Data Availability

The raw data from the proteomic analysis are available for review upon request to the corresponding author and will be submitted to appropriate data-sharing resources upon acceptance of the paper.

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
