# Peer review of "Chondroitin Sulfate Proteoglycan 4 Provides New Treatment Approach to Preventing Peritoneal Dissemination in Ovarian Cancer"

_ijms, 2024, doi:10.3390/ijms25031626_

Round 1

Reviewer 1 Report

Comments and Suggestions for Authors

Kaname Uno reviewed the effect of chondroitin sulfate proteoglycan 4 (CSPG4) in epithelial ovarian cancer (EOC). They found that CSPG4 was expressed in EOC and related with tumor aggressiveness, including proliferation, invasion, and sphere formation. I believe the results are of interest. However, there are several suggestions need to be addressed before publication.

Major revisions:

1. The authors found that CSPG4 was strongly stained of in the cell membrane of EOC samples. However, which kind of cells was strongly stained in the EOC samples, and why were HEY, ES2, A2780, and NOE used, what were the differences of these cell lines?

2. In the manuscript, CSPG4 was knockdown via siRNA in figure 2 and shRNA in Figure 3. Please explain the aim of these methods.

3. LEF1 induced strong expression of CSPG4 in HeLa cells via regulation of the promoter area of CSPG4. However, the mechanisms of CSPG4 regulate cell proliferation, migration, and invasion abilities in EOC cell lines were unknown. The author should discuss the molecular mechanisms in the part of Discussion.  

Minor comments:

1. In line 449, the authors should declare conflict of interest in the part of Conflicts of Interest.

2. In line 388, 1×106 GFP-labeled ES2 cells should be replaced by 1×106 GFP-labeled ES2 cells.

Comments on the Quality of English Language

Minor editing of English language required

Author Response

Kaname Uno reviewed the effect of chondroitin sulfate proteoglycan 4 (CSPG4) in epithelial ovarian cancer (EOC). They found that CSPG4 was expressed in EOC and related with tumor aggressiveness, including proliferation, invasion, and sphere formation. I believe the results are of interest. However, there are several suggestions need to be addressed before publication.

Response: Thank you very much for reviewing our manuscript and giving several constructive suggestions. Our response to your questions is below.

Major revisions:

  1. The authors found that CSPG4 was strongly stained of in the cell membrane of EOC samples. However, which kind of cells was strongly stained in the EOC samples, and why were HEY, ES2, A2780, and NOE used, what were the differences of these cell lines?

Response: We appreciate for our insufficient explanations. We have checked the qPCR level of CSPG4 in seven EOC cell lines (HEY, ES2, A2780, NOE, OV90, OVCAR3, and CaOV3) and compared to control cell lines (positive control was MDA-MB-231 and negative control was MCF7). As we showed in Figure. 1g and 1h, the mRNA levels of CSPG4 in HEY, ES2, A2780, and NOE were relatively the same level with the positive control cell line, MDA-MB-231. Moreover, we analyzed the protein level by Western blot in five cell lines (SKOV3, ES2, HEY, NOE, and OVCAR3). The results showed that protein expression correlated with mRNA expression, indicating that HEY, ES2, A2780, and NOE of the EOC cell lines express CSPG4. We highlighted this at Page 3, line 100-105.

The characteristics of these cell lines is considered to show more epithelial-mesenchymal transition (EMT) compared to other cell lines which was not expressed CSPG4. When CSPG4 was inhibited with shRNA in ES2, the morphology of the cells became round and showed more epithelial character as shown in Fig. 3a. Yang, et al. showed the relationship with CSPG4 and EMT character (Cancer Res. 2009). Previously, we showed that HEY, ES2, and NOE showed more EMT characters than other EOC cell lines (Int J Mol Sci. 2020). These results supports that the cell lines with strongly expressed CSPG4 show EMT characteristics. We highlighted this discussion at Page 14, in line 299-301.

  1. In the manuscript, CSPG4 was knockdown via siRNA in figure 2 and shRNA in Figure 3. Please explain the aim of these methods.

Response: As you mentioned, we created two different knockdown system via siRNA and shRNA. siRNA was used in several experiments to check proliferation and migration/invasion. These results showed the importance of CSPG4 in ovarian cancer cells as previous studies showed in melanoma and malignant mesothelioma. However, the efficacy of inhibition via siRNA is relatively short (up to 3 days). It is difficult to use siRNA knockdown system in vivo model. To assess the role of carcinogenesis using in vivo model for more than 1 week, we created shRNA knockdown system. Therefore, after Figure 3, we used shRNA knockdown system to inhibit CSPG4 expression. We have described several sentences to explain the aim of these methods (Page 3, line 130-131).

  1. LEF1 induced strong expression of CSPG4 in HeLa cells via regulation of the promoter area of CSPG4. However, the mechanisms of CSPG4 regulate cell proliferation, migration, and invasion abilities in EOC cell lines were unknown. The author should discuss the molecular mechanisms in the part of Discussion.

Response: We agree your constructive suggestions. In this study, we mainly focused on the mechanism of what factor regulated CPSG4 expression. And we revealed that LEF1 regulates CSPG4 expression. Previous studies showed several mechanisms of cell proliferation, migration, and invasion associated with CSPG4. We did not discuss about the mechanisms in our previous manuscript as you pointed out. According to your suggestion, we have added the mechanism in Discussion section with 7 references at Page 15, line 346-354, and line 356-358.

Minor comments:

  1. In line 449, the authors should declare conflict of interest in the part of Conflicts of Interest.

Response: We apologize for our mistake in the first submission. We do not have any conflicts of interest in this study. We have declared at the Conflict of Interest section in Page 19, line 565.

  1. In line 388, 1×106 GFP-labeled ES2 cells should be replaced by 1×106 GFP-labeled ES2 cells.

Response: We are sorry for our mistake in the first submission. We have changed it according to your suggestion.

Reviewer 2 Report

Comments and Suggestions for Authors

In this study, Kaname et al. demonstrated that CSPG4 is highly expressed in ovarian cancer and has prognostic values for patients' survival. In addition, the study also revealed a potential mechanism that contributes to this phenomenon. The findings may have clinical implications in the future. Several issues must be addressed before the paper can be published.

What is the expression level of CSPG4 in normal ovarian cells and tissues? The authors should include normal ovarian samples in Figure 1.

Please supplement the sample size for Figure 1D.

In Figure 1G, the labels for y are so weird. What do you mean by qPCR levels? It is better to use "the mRNA relative expression level."

Which cell line did the author use for the knockdown experiments? Please specify the name of the cell lines used for those experiments.

Please show the validated results of siRNA or shRNA knockdown efficacy. E.g., Western blot.

ES2 cells have a relatively low expression of CSPG4. Please indicate the rationales for why ES2 cells were selected for the study.

In Figure 5f, I cannot tell whether there is a metastasis only based on IVIS images. How can the author conclude that shCSPG4 has less metastasis? Please specify it.

Please avoid using "we first" or "it is the first time."

Materials and methods: please provide more patient information, including ages, tumor types, stages, etc.

For IHC, please indicate detailed reproducibility information, such as antibody concentration, the incubation temperature and time, the information on secondary antibodies, etc.

The author mentioned the orthotopic mouse model in Figure 5. Please include the details of orthotopic model establishment methods in this study.

Author Response

In this study, Kaname et al. demonstrated that CSPG4 is highly expressed in ovarian cancer and has prognostic values for patients' survival. In addition, the study also revealed a potential mechanism that contributes to this phenomenon. The findings may have clinical implications in the future. Several issues must be addressed before the paper can be published.

Response: Thank you very much for reviewing our manuscript and giving several constructive suggestions. Our response to your questions is below.

What is the expression level of CSPG4 in normal ovarian cells and tissues? The authors should include normal ovarian samples in Figure 1.

Response: Thank you very much for reasonable comments. We fully agree that we needed to check the CSPG4 expression in normal ovarian tissues, although it is difficult to collect normal ovaries. We have tried to reveal the expression of CSPG4 with several methods.

Firstly, we have observed CSPG4 expression with immunohistochemistry of our samples. We observed the samples which include both cancer and normal area. While cancer cells showed moderate expression of CSPG4, the normal epithelial and stromal area were negative or weak expression. We have added some figures of these difference in Supplementary Figure 1A and 1B. And we mentioned the results in the Results section at Page 2, line 90-91.

Secondly, we checked the mRNA relative expression level using public database at CSIOVDB: a microarray gene expression database of ovarian cancer subtype (http://csiovdb.mc.ntu.edu.tw/pages/CSIOVDB_CSPG4.html). The mRNA level in tumor tissues was significantly higher than that of ovarian surface epithelial (5.14 vs 4.95, p < 0.001). We have added this result in Result section (Page2, line 91-93) and Supplementary Figure 1C.

Lastly, we checked the protein expression in public data base at Protein Genome Atlas. Three kinds of antibodies for CSPG4 were used to stain three normal ovaries and revealed that normal ovarian epithelial and stroma cells were negative or low expression of CSPG4, although almost 80% of ovarian cancer tissues were positive. We have added this information in the Result section (Page 2, line 96-97).

Public data information was added in Material and Methods section at Page 16, line 417-425.

Please supplement the sample size for Figure 1D.

Response: We apologize for our insufficient explanation and thank you for pointing out that point. The first two was used for 12 patients, and the last antibody was used for 11 patients. We have added this information in the Figure 1D as (n = 12) and mentioned in Results section (Page 2, line 96).

In Figure 1G, the labels for y are so weird. What do you mean by qPCR levels? It is better to use "the mRNA relative expression level."

Response: We appreciate and agree your suggestion. We changed the expression to “The mRNA relative expression level” according to your advice.

Which cell line did the author use for the knockdown experiments? Please specify the name of the cell lines used for those experiments.

Response: We checked the expression of CSPG4 in several ovarian cancer cell lines. As we showed in Figure 1D and 1H, the mRNA expression level of ES2, A2780, and NOE was almost the similar to positive control cell line, MDA-MB-231. Among them, A2780 was difficult to use in invasion and migration assay. That is the reason why we chose ES2 and NOE cells as the appropriate cell lines for this experiment. We highlighted this at Page 3, line 100-105.

Please show the validated results of siRNA or shRNA knockdown efficacy. E.g., Western blot.

Response: Thank you for your kind indication. We showed the efficacy of siRNA with Western blot in Supplementary Figure 1A. And we have also checked the mRNA expression level with siRNA and reveal that the level of CSPG4 was significantly decreased using the siRNA. We have added this result at Page 3, line 109-111 and Supplementary Figure 1D and 1E.

To show the efficacy of shRNA, we showed decreased expression on the cell surface using FACS in Supplementary Figure 2A. And we have added the mRNA expression level of sh-negative and sh-CSPG4 as Supplementary Figure 2B and at Page 3, line 130-137.

ES2 cells have a relatively low expression of CSPG4. Please indicate the rationales for why ES2 cells were selected for the study.

Response: Thank you for your reasonable question, but the expression of CSPG4 in ES2 was not low compared to positive control MDA-MB-231. To check the relative expression level of CSPG4, we used the breast adenocarcinoma cell line MDA-MB-231 as positive control and MCF7 as negative control. The mRNA expression level was almost the same level of MDA-MB-231 as we showed in Figure 1G. Therefore, we regarded that ES2 was suitable cell line in this study to reveal the function of CSPG4. To reveal the reason, we have added this at Page 3, line 100-105.

In Figure 5f, I cannot tell whether there is a metastasis only based on IVIS images. How can the author conclude that shCSPG4 has less metastasis? Please specify it.

Response: We agree your opinion. While we revealed the decreased burden of metastasis to omentum and mesenchymal in Figure 5B-D, we could not conclude based on IVIS images. Therefore, we deleted the sentence “and did not show peritoneal metastasis” at Page4, line 181.

Please avoid using "we first" or "it is the first time."

Response: We agree your opinion. We have deleted “first” at Page 14, line 289, and changed the expression to “it is the first time” at Page 14, line 314.

Materials and methods: please provide more patient information, including ages, tumor types, stages, etc.

Response: We apologize for our insufficient explanation about clinical samples. We collected ovarian cancer tissues from January 2013 to December 2018. Among the period, we selected 52 samples because these samples had not only primary but also metastases resection. Eight samples of representative high-grade serous carcinoma were used for Western blot analysis. We have added this information in Material and Methods section (page 15, line 387-391).

For IHC, please indicate detailed reproducibility information, such as antibody concentration, the incubation temperature and time, the information on secondary antibodies, etc.

Response: We apologize for our insufficient explanation about immunohistochemistry. We have added several sentences to indicate detailed reproducibility information at Page 16, line 394-404.

The author mentioned the orthotopic mouse model in Figure 5. Please include the details of orthotopic model establishment methods in this study.

Response: We agree your suggestion. We have described the details of orthotopic model in Materials and Methods section at Page 18, line 527-534.

We have also added several methods to clarity using in this study.

Reviewer 3 Report

Comments and Suggestions for Authors

The manuscript by Kaname et al measured the expression of CSPG4 in ovarian cancer patients and investigated the roles of CSPG4 in ovarian cancer by knocking it down in ovarian cancer cell lines. They found that CSPG4 is associated with aggressive features of EOC and has the potential to be a therapeutic target. Some concerns are:

1.      Similar findings have been reported by another group in PMID: 34942534, which affects the novelty of this manuscript.

2.      Missing the description of some important methods, such as IF, measurement of movement on a single layer of mesothelial cells.

3.      The use of ‘Ascites’ is wrong, it should be ‘peritoneal wash’ rather than ‘ascites’ according to the author’s description.

4.      Fig.1g lacks a normal control.

5.      Fig. 1h shows HEY expresses a high level of CSPG4, which makes it a better study model than other cells. Why didn’t the author use it?

6.      The use of HELA cells in reporter assay is very confusing. Why didn’t the author use ovarian cancer cell lines?

Comments on the Quality of English Language

The manuscript needs some language editing.

Author Response

The manuscript by Kaname et al measured the expression of CSPG4 in ovarian cancer patients and investigated the roles of CSPG4 in ovarian cancer by knocking it down in ovarian cancer cell lines. They found that CSPG4 is associated with aggressive features of EOC and has the potential to be a therapeutic target. Some concerns are:

Thank you very much for reviewing our manuscript and giving several constructive suggestions. Our response to your questions is below.

  1. Similar findings have been reported by another group in PMID: 34942534, which affects the novelty of this manuscript.

Response: Thank you very much for your suggestion. We agree that to show our novelty compared to similar report is necessary. We referred the suggested study as ref. 19. Our study has at least three novelties compared to the study.

Firstly, the method of spheroid formation was different although the previous study also mentioned that CSPG4 was related to spheroid formation. We continuously observed spheroid formation and how EOC cells formed spheroids when CSPG4 was inhibited. On the other hand, the authors of the previous study only counted the number of spheroids on a dish at one time point regardless of size and aggregation. Moreover, we assessed the spheroid formation in peritoneal wash fluid in vivo model and revealed that the spheroid formation was related to induce peritoneal dissemination. Our experimental model is closer to clinical situations and has the novelty that CSPG4 is related to spheroid formation and survival in ascites. We have added this comparing to the suggested study at Page 14, line332-336.

Secondly, we revealed that interaction between ovarian cancer cells and mesothelial cells via CSPG4-FN1. CSPG4 has extracellular domain which bind to integrin and fibronectin. In this study, we suggested that ovarian cancer cells could attach and move on the mesothelial surface with CSPG4-FN1 interaction. These results strengthen and have novelties that CSPG4 is an important molecule to attach and induce peritoneal metastasis in ovarian cancer. We highlighted this point at Page 14, line 322-324.

Finally, we focused on the regulation of CSPG4 and revealed that LEF1 regulate the expression of CSPG4. The regulatory mechanism of CSPG4 has not fully understood, and PAX3 is reported to regulate CSPG4 expression. However, in ovarian cancer, we showed that inhibition of PAX3 did not decrease CSPG4 expression. Instead, in the upper stream of CSPG4, there were several LEF1 binding regions and inhibition of LEF1 decreased the expression of CSPG4. This is the first report that LEF1 is a key molecule to regulate CSPG4 expression. The previous study did not perform any regulatory mechanisms about CSPG4. This regulatory mechanism is also our novelty in this study. To emphasize these points, we added several sentences in Discussion section comparing the previous study at Page 15, line 348-360.

  1. Missing the description of some important methods, such as IF, measurement of movement on a single layer of mesothelial cells.

Response: Thank you for your kind suggestion. We fully agree your opinion. At the first submission, we mistakenly might not upload supplementary Material and Methods file. Therefore, our previous manuscript lack of several important methods. We have described with respect to the details of some methods in "Materials and Methods".

“Western blot analysis” at Page 16, line 411-415.

“Public data analysis” at Page 16, line 417-424.

“Quantitative real-time PCR (qPCR)” at Page 16, line425-434.

“Transwell migration and invasion assays” at Page 17, line 464-472.

“Wound healing” at Page 17, line 473-482.

“Proteomic analysis” at Page17, lime 483-489.

“Immunofluorescence” at Page 17, line 490-496.

“Mesothelial migration assay” at Page 18, line 505-512.

“Analysis of cell surface markers” at Page 18, line 535-540.

We have also added information in Patient sample collection, Cell lines, Knockdown CSPG4 expression, and Mouse model studies.

  1. The use of ‘Ascites’ is wrong, it should be ‘peritoneal wash’ rather than ‘ascites’ according to the author’s description.

Response: We agree your opinion. We changed the expression to “peritoneal wash fluid” or “the fluid” at Page 4, line 177-181, Page 14, line 332-336, Page 18, line 519-522 and Figure legend of the Figure 5. We have also changed the expression in the sentence in Abstract to “In the peritoneal wash fluid from mice injected with sh-CSPG4 EOC cells, significantly fewer spheroids were present.”

  1. Fig.1g lacks a normal control.

Response: Thank you for your reasonable question. As we answered the comments from the reviewer 2, we agree this information is necessary to show. However, it is very difficult to collect normal ovary from patients. Moreover, normal ovary contains not only epithelial surface but also stroma and follicles. We performed three methods to reveal the expression of CSPG4 in normal ovaries. Below are our responses.

Firstly, we have observed CSPG4 expression with immunohistochemistry of our samples. We observed the samples which include both cancer and normal area. While cancer cells showed moderate expression of CSPG4, the normal epithelial and stromal area were negative or weak expression. We have added some figures of these difference in Supplementary Figure 1A and 1B. And we mentioned the results in the Results section at Page 2, line 90-91.

Secondly, we checked the mRNA relative expression level using public database at CSIOVDB: a microarray gene expression database of ovarian cancer subtype (http://csiovdb.mc.ntu.edu.tw/pages/CSIOVDB_CSPG4.html). The mRNA level in tumor tissues was significantly higher than that of ovarian surface epithelial (5.14 vs 4.95, p < 0.001). We have added this result in Result section (Page2, line 91-93) and Supplementary Figure 1C.

Lastly, we checked the protein expression in public data base at Protein Genome Atlas. Three kinds of antibodies of CSPG4 were used to stain three normal ovaries and revealed that normal ovarian epithelial and stroma cells were negative or low expression of CSPG4, although almost 80% of ovarian cancer tissues were positive. We have added this information in the Result section (Page 2, line 96-97).

  1. Fig. 1h shows HEY expresses a high level of CSPG4, which makes it a better study model than other cells. Why didn’t the author use it?

Response: We can understand your question and suggestion. HEY strongly expressed CSPG4 even compared to positive control cell line. We regarded HEY was not suitable in our proposed experiments because the characteristics of the cell line is very different from ovarian cancer and several siRNA and shRNA of CSPG4 could not knockdown properly.

We agree that we need to specify this. We have added one sentence to clarify the cell lines at Page 3, line 100-105.

  1. The use of HELA cells in reporter assay is very confusing. Why didn’t the author use ovarian cancer cell lines?

Response: Thank you for your kind indication. HEK293 and HeLa cells demonstrate high transfection efficiency for many genes, allowing for effective introduction of foreign genes. Therefore, these cells are often used for luciferase assays that require simultaneous transfection of multiple plasmids. We also usually use HEK293 cells or HeLa cells when performing luciferase assay. In this study, we performed luciferase assay using HEK293 cells, but it did not work well, so we used the analysis results obtained with HeLa cells. Not to be confusing for readers, we have added this reason at 18, line 542-543.

Reviewer 4 Report

Comments and Suggestions for Authors

General comment:

Kaname et al describe the importance of CSPG4 for peritoneum metastasis EOC patients, potentially regulated by LEF1. First, they found the around 40% of EOC expressing CSPG4, which is relating to poor prognosis. The in vitro study found a consistent role in promoting migration and invasion, which is comparable to other report. Using sphere forming assay and in vivo metastasis/ascites observation, they claim CSPG4 is critical for peritoneum metastasis. At last, they found CSPG4 is upregulated by LEF1, instead of PAX6. In general, this article is fine to be published since it contains new data to the field. Some minor criticism needed to be revised.

Detailed critique:

1.     In the Figure 1, the patient demography is important for readers to understand what exactly the cohort looks like. Please add the EOC patient demography. What’s the major CSPG4 expressed EOC subtype is important information to realize what exactly CSPG4 play role in EOC disease.

2.     The cell line profile of CSPG4 (Fig. 1g) is good for readers to reference the basal CSPG4 expression in EOC cells. However, why author knockdown ES1, instead of Hey8 cells to test cellular functions? Although author stated in figure legend and result section that MDA-MB-231 is used for positive control, it’s still requiring proper citation.

3.     The data using siCSPG and anti-CSPG4 antibody to suppress cellular phenotype in vitro is interesting experiment (Fig. 2). Then, they used shCSPG4 to perform the similar experiment again (Fig. 3). It seems redundant to reviewer. Although the authors described cellular morphology changed in the shCSPG experiments, it raised more curiosity that the discrepancy between siRNA or shRNA might be reagent bias, instead of true phenotype of CSPG4. Author may discuss that and proposed best explanation to this curiosity.

4.     The mechanism for CSPG4 in peritoneum metastasis was suspected to be FN1 and TGFb1 related, based on their discuss section. It’s good to discuss the potential mechanism, yet, put in the conclusion cartoon is another issue. Reviewer suggest author to restrain the claim area in their data, but not extend too much outside this study.

Author Response

Kaname et al describe the importance of CSPG4 for peritoneum metastasis EOC patients, potentially regulated by LEF1. First, they found the around 40% of EOC expressing CSPG4, which is relating to poor prognosis. The in vitro study found a consistent role in promoting migration and invasion, which is comparable to other report. Using sphere forming assay and in vivo metastasis/ascites observation, they claim CSPG4 is critical for peritoneum metastasis. At last, they found CSPG4 is upregulated by LEF1, instead of PAX6. In general, this article is fine to be published since it contains new data to the field. Some minor criticism needed to be revised.

Response: We thank the reviewer for reviewing and summarizing our manuscript and giving us these appreciative words. Our response to your questions are below.

  1. In the Figure 1, the patient demography is important for readers to understand what exactly the cohort looks like. Please add the EOC patient demography. What’s the major CSPG4 expressed EOC subtype is important information to realize what exactly CSPG4 play role in EOC disease.

Response: Thank you for your important suggestions. We agree the demography is important information for readers. We included the included patient’s information. Our included patients were all Japanese women. We included the cases which have both primary and metastasis samples regardless of histology. Therefore, most of the patients were diagnosed with high-grade serous except for some patients with clear cell carcinoma. The number of patients were not enough to analyze depend on the different histology. The patient’s information were added at Page 15, line 389-394. We agree that the lack of information related to subtype can be a limitation. We have mentioned this point in limitation at Page 15, line 374-376.

  1. The cell line profile of CSPG4 (Fig. 1g) is good for readers to reference the basal CSPG4 expression in EOC cells. However, why author knockdown ES1, instead of Hey8 cells to test cellular functions? Although author stated in figure legend and result section that MDA-MB-231 is used for positive control, it’s still requiring proper citation.

Response: Thank you for your constructive criticism. As you mentioned, we used the breast cancer cell line MDA-MB-231 as positive control of CSPG4 expression. This is because Amoury et al showed MDA-MB-231 expressed CSPG4 and suggested the relationship between CSPG4 and BRCA1 mutation (Amoury et al. IJC. 2016). Therefore, we regarded this cell line could be available as positive control of ovarian cancer cell. As you pointed out, we need to highlight this information. We have described at Page 3, line 101-102 and Page16-17, line 447-449 with proper references.

When the mRNA expression level of CSPG4 in several ovarian cancer cell lines was compared to the positive control, MDA-MB-231, ES2, A2780, and NOE showed almost similar mRNA expression while HEY was very high. We regarded HEY was not suitable in our proposed experiments because the characteristics of the cell line is very different from ovarian cancer and several siRNA and shRNA of CSPG4 could not knockdown properly.

We agree that we need to specify this. We have added one sentence to clarify the cell lines at Page 3, line 100-105.

  1. The data using siCSPG and anti-CSPG4 antibody to suppress cellular phenotype in vitro is interesting experiment (Fig. 2). Then, they used shCSPG4 to perform the similar experiment again (Fig. 3). It seems redundant to reviewer. Although the authors described cellular morphology changed in the shCSPG experiments, it raised more curiosity that the discrepancy between siRNA or shRNA might be reagent bias, instead of true phenotype of CSPG4. Author may discuss that and proposed best explanation to this curiosity.

Response: Thank you for your reasonable suggestions. In the beginning, we tried to reveal the importance of CSPG4 in ovarian cancer via siRNA. And as you mentioned from the in vitro study revealed a consistent role in promoting migration and invasion as previous studies showed in melanoma and malignant mesothelioma. However, the efficacy of inhibition via siRNA is relatively short (up to 3 days). It is difficult to use siRNA knockdown system in vivo model. To assess the role of carcinogenesis and spheroid formation using in vivo model for more than 1 week, we created shRNA knockdown system. We believe that getting similar results using both siRNA and shRNA could be important, although a little redundant. Therefore, after Figure 3, we used shRNA knockdown system to inhibit CSPG4 expression. We have described several sentences to explain the aim of these methods (Page 3, line 130-131).

  1. The mechanism for CSPG4 in peritoneum metastasis was suspected to be FN1 and TGFb1 related, based on their discuss section. It’s good to discuss the potential mechanism, yet, put in the conclusion cartoon is another issue. Reviewer suggest author to restrain the claim area in their data, but not extend too much outside this study.

Response: We agree your reasonable suggestions and thank the reviewer for constructive criticism. In this study, we focused on how EOC cells attached on the surface of peritoneal membrane covered by a single layer of mesothelial cells. CSPG4 has extracellular domain which bind to integrin and fibronectin. We suggested that ovarian cancer cells could attach and move on the mesothelial surface with CSPG4-FN1 interaction using mesothelial migration assay. To reveal the interaction between EOC cells and mesothelial cell via CSPG4-FN1 binding is considered a new data. But as you suggested, these data could not be enough to extend. We suggested this mechanism in Discussion section at Page 14, line 322-324. Not to extend this too much, we put quotation mark in the figure legend in Figure 6 according to your advice.

Round 2

Reviewer 2 Report

Comments and Suggestions for Authors

Thank you for the responses. I have no further questions.

Author Response

Thank you for the responses. I have no further questions.

Response: Thank you very much for taking your time for reviewing our manuscript. We fully appreciate your constructive suggestions to improve our manuscript.

Reviewer 3 Report

Comments and Suggestions for Authors

ChIP assay using EOC cells is necessary for supporting the conclusion 'LEF1 is a key regulator of CSPG4 expres-195 sion in EOC cells'.

Comments on the Quality of English Language

No comments.

Author Response

ChIP assay using EOC cells is necessary for supporting the conclusion 'LEF1 is a key regulator of CSPG4 expression in EOC cells'.

Response: Thank you for your suggestion. To reveal the relation between LEF1 and CSPG4, we have performed luciferase assay using HeLa cells. We think that this model is the simplest and reasonable method to reveal the relation between LEF1 and promotor region of CSPG4 (below figure). And, we used HeLa cells because these cells have high transfection efficiencies and are widely used in the experiments to reveal a regulator of specific protein expression. But we fully understand your question that we also need to reveal this relation even in ovarian cancer cell lines.

In this revision, we used ovarian cancer cell line (ES2) according to your previous suggestion. ES2 has endogenous LEF1 expression. The luciferase activity was significantly lower when LEF1 was inhibited in ES2 cells. This result indicates that the promoter of CSPG4 was dependent to the level of LEF1. It has been added this result as Figure 6F in our revised manuscript (Page 4-5, line 196-207).

As you mentioned, ChIP assay is also reasonable method to reveal the relationship between LEF1 and the promotor region of CSPG4. Unfortunately, we have never performed this method. Although we start preparing the method, it will take more than one month that some materials will be delivered due to Christmas and New Year holidays. Therefore, we proceeded the luciferase assay using EOC cell lines to strengthen our conclusion.

And we agree that we should change the sentence “LEF1 is a key regulator of CSPG4 expression in EOC cells” to “LEF1 could be a key regulator of CSPG4 expression in EOC cell” (Page 5, line 208). We hope you will agree with us that the paper has improved substantially.
